# Numerical Simulation of the Effects of Scanning Strategies on the Aluminum Evaporation of Titanium Alloy in the Electron Beam Cold Hearth Melting Process

**DOI:** 10.3390/ma15030820

**Published:** 2022-01-21

**Authors:** Van-Doi Truong, Yong-Taek Hyun, Jong Woo Won, Wonjoo Lee, Jonghun Yoon

**Affiliations:** 1Department of Mechanical Design Engineering, Hanyang University, 222 Wangsimni-ro, Seongdonggu, Seoul 04763, Korea; vandoitruong1997@gmail.com (V.-D.T.); wj6478@gmail.com (W.L.); 2BK21 FOUR ERICA-ACE Center, Hanyang University, 55 Hanyangdaehak-ro, Sangnok-gu, Ansan-si 15588, Gyeonggi-do, Korea; 3Department of Titanium, Korea Institute of Materials Science, 797 Changwondaero, Seongsan-gu, Changwon-si 51508, Gyeongsangnam-do, Korea; ythyun@kims.re.kr (Y.-T.H.); jwwon@kims.re.kr (J.W.W.); 4Department of Mechanical Engineering, Hanyang University, 55 Hanyangdaehak-ro, Sangnok-gu, Ansan-si 15588, Gyeonggi-do, Korea; 5AIDICOME Inc., 55 Hanyangdaehak-ro, Sangnok-gu, Ansan-si 15588, Gyeonggi-do, Korea

**Keywords:** EBCHM process, aluminum evaporation, scanning strategies

## Abstract

In the production of titanium alloy, the electron beam cold hearth melting (EBCHM) process is commonly used due to its effectiveness and the high quality of the end product. However, its main drawback is the significant loss of elements such as aluminum (Al) due to evaporation under the vacuum environment. Numerical coupled thermal-flow models were here developed to investigate the effects of scanning strategies on Al loss in a titanium alloy during EBCHM. The validation model was successful in comparison with previously published experimental data. The Al mass fraction results at the outlet of the water-cooled hearth were strongly influenced by changes in the applied scanning strategies. The results indicated that the Al mass fraction loss could be reduced by using the full-hearth scanning strategies.

## 1. Introduction

Titanium and its alloys are widely used in high-performance products in a variety of industries, including the aerospace, automotive, biomedical, and energy industries, due to their outstanding properties such as light weight, excellent strength, high melting temperature, and exceptional corrosion resistance [1,2]. Hence, requirements for precise quality control are critical in the manufacturing processes of titanium alloys. There are currently two consolidation processes used to produce commercial titanium alloy ingots: vacuum arc remelting (VAR) and cold hearth melting (CHM) [2,3]. CHM is a recently developed method that has several advantages over VAR, such as the automatic separation of high-density inclusions, better control of the heat flux, flexibility in the choice of the feed material, and the fact that it allows the direct casting of non-axisymmetric shapes [1,2,4]. There are two types of CHM furnace with different heat sources: plasma arc melting (PAM) and electron beam cold hearth melting (EBCHM). An EBCHM furnace operates in a vacuum, whereas a PAM furnace operates under the partial pressure of argon [2], giving it the ability to eliminate volatile elements such as chloride and hydrogen in the hearth [4]. Given these advantages, EBCHM is currently dominant in the production of premium-quality titanium alloys.

A schematic of the EBCHM process is given in Figure 1, showing its two stages: (1) melting and refining and (2) ingot casting or solidification. First, the feed material (such as sponge or scrab obtained from the conversion of ore with abundant titanium into metallic titanium) is fed into the hearth. The feed materials are melted by electron beam guns and then the molten metal flows along the water-cooled copper melting hearth. Its temperature is continuously maintained by another computationally controlled electron beam gun on the top of the liquid surface to compensate for the heat from radiation and conduction. In this refining stage, any high-density inclusions sink and are trapped at the bottom of the crucible, and the low-density inclusions evaporate or are dissolved in the liquid metal. Finally, the molten metal flows over the outlet lip into a water-cooled copper mold to solidify and is withdrawn in a solid form such as a round or slab-shaped ingot.

The main challenge of this process is control over the ingot chemistry due to the significant losses of elements in the titanium alloy under high vapor pressure, such as aluminum (Al) lost by evaporation under the vacuum environment in the hearth. Several studies have sought to improve the technology in the last two decades [4,5,6,7,8,9,10,11]. Isawa et al. [7] investigated Al evaporation behavior in the EBCHM process using a 250 kW electron beam (EB) furnace. Their results showed that Al evaporation occurred mainly in the hearth and the Al yield (which is the ratio of the final Al concentration to the initial concentration) increased with the casting rate. Based on their experimental results, the rate equation of Al evaporation was determined by taking account of both reactions at the interface and mass transfer in the melt. A recent study by Zhang et al. [11] experimentally examined the relationships between the melting rate and the chemical composition of a Ti-6Al-4V alloy. Their results showed that at the melting rates of 70, 100, and 140 kg/h, the Al volatilization losses of the melted ingots were 22%, 16%, and 12%, respectively. Siwiec et al. [10] used a pilot-scale experiment to analyze the Al loss due to evaporation in a vacuum induction furnace at 5 to 1000 Pa and at 1973 K, 1998 K, and 2023 K. Their results confirmed a ~14% to ~28% Al loss from the alloy compared to the initial composition (5.5% of the initial mass). Powell et al. [8,9] estimated the activity coefficient of Al in a titanium alloy to be 0.063 at temperatures between 1943 K and 2432 K based on the melt and condensed vapor composition and the hearth temperature profile. Additionally, Choi et al. [6] estimated the activity coefficient at 2133 K to be in the range from 0.044 to 0.0495 in two different experiment approaches using a laboratory electron beam furnace.

These studies provided influence trends for several related parameters, such as the initial concentration, melting rate, and heat source power; however, there are obvious limitations with regard to direct measurement and high costs, making it difficult to produce a comprehensive analysis with those methods. Thanks to the rapid development of computation, however, more recent studies tend to use mathematical models to simulate and investigate this process. Powell et al. [8,9] constructed a mathematical model to investigate the evaporation rate with multiple components. Their mathematical model showed that the evaporation rate increased very quickly as the power increased, making it easier to dampen out changes in the composition without generating transient flow near the beam. Bellot et al. [4] constructed a simulation to investigate the influence of the casting rate and the beam scanning frequency on the evaporation of Al in both a pilot-scale and an industrial-type furnace. They showed that the casting rate had a weak effect on the volatilization flux, but a high casting rate reduced the Al losses of the remelted charge. In terms of bath scanning frequency, there was a 10% decrease in the Al loss when it increased from 0.5 to 10 Hz; moreover, a quasi-stationary state was reached when the frequency was higher than 10 Hz.

Akhonin et al. [5] presented a mathematical model to describe the kinetics of Al evaporation in the EBCHM process. Their model could couple the mass and energy balance to predict the influence of the process parameters (melting rate and power input) and the compositions of the charge on the final ingot chemistry. Gao’s studies [12,13] focused only on the solidification stages, using a three-dimensional, transient, multiphasic, numerical model to investigate evaporation, pool evolution, and solidification. These studies demonstrated the effect of the casting rate and suggested that the correct inlet-flow velocity and temperature might enhance composition homogenization. Many of the mentioned studies paid attention to specific stages of the EBCHM process—either the refining hearth or the solidification stage—in order to enhance the ingot quality by minimizing losses of elements in the vacuum. However, to date there has been no investigation of the scanning strategy used by the electron beam on the free liquid surface of the hearth, where most of the element losses actually occur.

In this study, a coupled thermal-flow model was constructed to investigate the effects of scanning strategies on the Al evaporation rate of Ti-6Al-4V in the EBCHM process. Various scanning strategy configurations with constant casting rates, power levels, and input concentrations were also applied to obtain a comprehensive view of the related factors that can affect Al losses in the EBCHM process. The simple coupled thermal-flow model was validated using the reference experimental data in [14] to demonstrate its potential in industrial practices. Some recommendations related to scanning strategies that may help optimize and enhance ingot quality are also given.

## 2. Materials and Methods

The numerical model was designed to investigate the effects of scanning strategies on Al loss and to develop recommendations for optimizing the Al loss in the EBCHM process. The entire model was constructed and simulated using the commercial CFD software Ansys Fluent v20.1. It assumed that the electron beam heat source was applied on a free, flat, liquid surface, and the evaporation rate was computed using Langmuir’s theory.

### 2.1. Governing Equations for the EBCHM Process

The EBCHM process is complex due to various physical phenomena related to heat transfer and fluid flow. When the heat flux is absorbed to maintain the liquid phase, the heat loss due to heat radiation on the refining pool surface and the heat conduction through the water-cooled hearth wall are the main mechanisms. The surface tension represents the gradient of the temperature effect on the melting pool. The heat convection into the ambient atmosphere and the heat loss due to the titanium alloy element evaporation were ignored here. Due to the low velocity of the fluid flow during this process, a Newtonian, laminar, and incompressible flow was assumed for simplicity in this model. Therefore, a set of well-known transport equations were used to fully describe the coupled thermal-fluid model in the EBCHM process, including the mass, momentum, heat, and solute conservation. The mathematic details of these equations are described in the Fluent Theory Guide [15], and only the final equations in the three-dimensional Cartesian coordinate system are presented in this section.

The continuity equations describing the mass conservation in the hearth can be written as:(1)$$\nabla \cdot \left(\rho \vec{v}\right)=0$$
where ρ is the density of the material, t is time, and $\vec{v}$ is the vector of the fluid velocity.

The momentum conservation equation describing the incompressible melting flow under gravity is given as:(2)$$\frac{\partial }{\partial t}\left(\rho \vec{v}\right)+\nabla \cdot \left(\rho \vec{v}\vec{v}\right)=-\nabla p+\nabla \cdot \left(\overset{=}{\tau }\right)+\rho \vec{g}$$
in which the stress tensor in vector form is:(3)$$\overset{=}{\tau }=\mu \left[\left(\nabla \vec{v}+\nabla {\vec{v}}^{T}\right)-\frac{2}{3}\nabla \vec{v}I\right]$$
where p is the static pressure, μ is the dynamic viscosity, and g is the gravitational acceleration. As mentioned before, since a low Reynolds number could be estimated, the turbulence model was assumed to be a laminar model.

The energy conservation equation representing the heat transfer model can be written as:(4)$$\frac{\partial }{\partial t}\left(\rho H\right)+\nabla \cdot \left(\rho \vec{v}H\right)=\nabla \cdot \left(k\nabla T\right)+S$$
where T is the temperature, H is the specific heat, k is the thermal conductivity, and S is the heat source term.

During the EBCHM process, the liquid fraction is tracked from 0 (for a solid) to 1 (for a liquid) due to the solidification process. The enthalpy of the material can be written as:(5)H=h+∆H
in which ∆H is the latent heat; h is the sensible enthalpy, with $h=h_{ref}+\int_{T_{ref}}^{R}C_{p}dT$; and $C_{p},\;h_{ref},\;\mathrm{and}\;T_{ref}$ are the specific heat at constant pressure, the reference enthalpy, and the reference temperature, respectively.

The liquid fraction in the molten pool can be defined as:(6)$$\begin{array}{c} \beta =0\;\;\;\;\;\;if\;T<T_{solidus}\\ \;\beta =1\;\;\;\;\;\;if\;T>T_{liquidus}\\ \beta =\frac{T-T_{solidus}}{T_{liquidus}-T_{solidus}}\;if\;T_{solidus}<T<T_{liquidus} \end{array}$$

An additional transport equation used to define the concentration of the Al mass fraction can be expressed as:(7)$$\frac{\partial \rho \phi }{\partial t}+\frac{\partial }{\partial x_{i}}\left(\rho u_{i}\phi -\Gamma_{k}\frac{\partial \phi }{\partial x_{i}}\right)=0$$
where ϕ represents the composition of the Al.

### 2.2. Computational Domain and Boundary Conditions

As the free fluid surface on the hearth has the largest area under the exposure of the electron beam’s power, it was assumed to be a main evaporation site where the relatively large evaporation reaction area and the long residence time of the fluid flow are primary. Hence, considering only the refining stage, the geometry of the numerical domain is shown in Figure 2, in which a three-dimensional rectangular hearth with a length of 1000 mm and a width of 500 mm illustrates an industrial-scale application [4]. Due to the high calculation cost for a full numerical model with a fine grid, only a half-symmetry computational domain was chosen to improve the efficiency. In this study, the structural hexahedral grid was generated using a multi-zone method. An element size of 5 mm and total number of elements of 104,900 were used to obtain grid-independent converged solutions. The SIMPLE algorithm was applied for pressure–velocity coupling and the least squares cell-based method was used to calculate gradient. The discretization model for pressure was second order and the second order upwind was used for momentum, energy, and user-defined scalars. A constant of 10,000 was used for the mushy zone parameter to capture the solid–liquid interface evolution. The time-step size for quasi-stationary state calculation was 0.01 s after using the time-averaged heat-flux approach.

In terms of boundary conditions, four regions were used to describe the fluid flow—thermal and solute—and they were divided across the surface of the computational domain. These regions comprised the inlet area, the top surface of the hearth, the hearth walls, and the outlet area.

The inlet and outlet geometries were assumed to be trapezoid regions with widths of 100 mm and 20 mm, respectively, a height of 20 mm, and a slope of 1. The titanium alloy was initially assumed to be melted into a liquid at a temperature of 1700 °C, and it flowed into the inlet at a constant flow rate of 400 kg/h.

On the top of the melting pool, several boundary conditions were applied to describe the heat transfer, evaporation, and surface tension. The combined power, $P_{EB}$, applied in this study was 800 kW, assuming an effective heat of 70% (refer to the scanning strategy details presented in Section 2.3). The roof and the wall of the vacuum chamber and the copper mold were assumed to behave as gray bodies. Therefore, only the loss of heat flux due to radiation was considered in the following equation, and the heat loss from the hearth due to evaporation was assumed to be negligible.
(8)$$q_{rad}=\varepsilon \sigma_{sb}\left(T^{4}-T_{\infty }^{4}\right)$$

For the Al concentration, the evaporation flux of the Al can be calculated based on Langmuir’s law:(9)$${\dot{m}}_{Al}=\gamma_{Al}X_{Al}P_{Al}^{0}\sqrt{\frac{M_{Al}}{2\pi RT}}$$
where the activity coefficient $\gamma_{Al}$ applied in this study was 0.044 [6]. The vapor pressure of the Al can be estimated using the Clausius–Clapeyron equation [16]:(10)$$P_{Al}^{0}=133.322\times 10^{-\frac{16450}{T}+12.3+\log T}$$

The mole fraction of the Al can be calculated as follows:(11)$$X_{Al}=\frac{w_{Al}/M_{Al}}{w_{Al}/M_{Al}+w_{Ti}/M_{Ti}+w_{V}/M_{V}}$$

The Marangoni force is caused by variations in the temperature and composition of the liquid surface. Therefore, the net force acts from a low-surface-tension area to a high-surface-tension area. The surface tension equation can be written as:(12)$$F_{M}\left(T,C\right)=\frac{\partial \sigma }{\partial T}\frac{\partial T}{\partial n}+\frac{\partial \sigma }{\partial C}\frac{\partial C}{\partial n}$$
where ∂σ/∂T and ∂σ/∂C indicate the surface tension due to the temperature and the composition, respectively.

At the wall of the crucible, only the thermal condition was applied to express the heat transfer between the solidified metal and the water-cooled walls. The Fourier condition is given in this equation, and the heat transfer coefficient was considered to be $375\;W/m^{2}/K$ [4]:(13)$$q_{con}=h\left(T-T_{mold}\right)$$

The material properties of the Ti-6Al-4V alloy with which the effects of electron beam scanning strategies were investigated are shown in Table 1 [4].

### 2.3. The Electron Beam Scanning Strategies

The evaporation rate of the alloy elements, which are extremely sensitive to temperature fluctuation, is strong and is a highly nonlinear function of temperature [8]. Therefore, two approaches that utilized various case studies were developed here to investigate the scanning strategies’ effects. First, the constant electron beam power, which was assumed to be a uniform heat flux, was applied on the surface in three specific areas (cases 1, 2, and 3 in Figure 3). The heat flux can be written as:(14)$$q_{EB}\left(x,y\right)=\frac{\eta_{EB}P_{EB}}{A}$$
where $\eta_{EB}$ is the absorption coefficient, $P_{EB}$ is the electron beam power, and A is the total heat flux area.

Second, the electron beam heat flux was considered as a Gaussian distribution. In this context, the elliptical pattern used in most industrial practices [4] was applied (case 4 in Figure 3). Due to the high frequency, a time-averaged heat-flux approach was applied by summing the heat flux from each of N points in the profile. The heat flux absorbed at an arbitrary point (*x*, *y*) at a certain time (*t*) can be expressed as:(15)$$q_{EB}\left(x,y\right)=\eta_{EB}P_{EB}\frac{1}{2\pi \sigma^{2}}\;\overset{N}{\underset{i=1}{\sum }}\frac{t_{i}}{t_{N}}e^{-\frac{{\left(x-x_{EB}\left(t\right)\right)}^{2}+{\left(y-y_{EB}\left(t\right)\right)}^{2}}{2\sigma^{2}}}$$
where $x_{EB}\;\mathrm{and}\;y_{EB}$ indicate the center position of the electron beam, σ is the electron beam radius, and $t_{i}\;\mathrm{and}\;t_{N}$ are the dwell time at each point and the total time required to complete the whole pattern, respectively. The electron beam scanning parameters are shown in Table 2.

## 3. Validation of the Numerical Model

### 3.1. Validation of Coupled Thermal-Flow Model

To quantify the accuracy of the correspondence between reality and the simulation at an industrial scale, the influence of the flow and pool conditions on the Al evaporation had to be validated. However, due to the very high cost of conducting such experiments at the industrial scale, the experiments were implemented at the pilot scale. Since evaporation is usually associated with the thin liquid layer on the top surface in the hearth, a simple coupled thermal-flow button model (an electron beam button melting (EBBM) model) was built and verified with the experimental data from Zhang’s publication [14]. The geometry and boundary conditions of the two-dimensional axisymmetric model of the button are shown in Figure 4 and Figure 5.

The Ti-6Al-4V material properties shown in Table 1 were applied to this domain. The left-side boundary was the axis where there was no mass or heat transfer. The right side touched the water-cooled mold, so the heat loss due to conduction was considered. The bottom side had no physical contact with the water-cooled mold, so radiation was assumed to be the primary heat transfer mechanism. On the top surface, the heat flux distribution was applied, and the surface tension was related to the temperature and the Al concentration gradient. The evaporation of the Al element was also included to validate the Al mass fraction with the experimental results. Some main parameters are shown in Table 3 and detailed information on the experimental setup and results can be found in [14].

### 3.2. Validation Model Results

As mentioned before, during the cooling process the thermal field was cooled under conduction and radiation. Figure 6 shows the temperature variation at three locations in the EBBM sample, demonstrating that the temperature trends of the proposed model and the experiment data were similar. The temperature at the three locations gradually decreased and reached almost 600 °C after 300 s of cooling. Figure 7 compares the Al concentration profiles at both the top surface (Figure 7a) and the centerline (Figure 7b) of the proposed model, as well as the measured data from experiments described in [14]. The results match very well and demonstrate that the proposed model could illustrate the experiment process. Apparently, the significant reductions in the Al concentration at the radius, from 0 to 30 mm (Figure 7a), and at the centerline depth, from 0 to 15 mm (Figure 7b), were due to evaporation during the melting process.

## 4. Results and Discussion

### 4.1. Temperature Distribution and Melt Pool Profile under Different Scanning Strategies

In order to understand the interaction between the electron beam and liquid flow on the free surface and the resulting melt flow behavior during EBCHM, four strategies for scanning over the top surface of the refining pool were deployed and simulated. In industrial practice, because the EB frequency is very high (1 kHz) [4], it is very time consuming to capture the moving heat source in a macro-scale simulation. The mentioned time-averaged heat-flux approach demonstrated its potential when integrated into the model. Instead of calculating the heat flux at every, very small, time-step size based on the location of the EB center, the total heat flux absorbed at an arbitrary point at a particular time was pre-computed with four different scanning strategies. Therefore, the fixed EB heat flux over the top surface of each case illustrated in Figure 3 was utilized to save calculation time. Furthermore, the deviation in the temperature distribution depends not only on the input heat flux but also on the radiation and the influence of fluid flow.

Figure 8a shows the temperature distribution for the top surface of the refining pool after 550 s with the four different scanning strategies presented in Section 2.3. In all four cases, temperatures over 2000 K were reached. The temperature distribution was almost uniform in case 1 and there was less fluctuation in case 4, while the higher temperatures occurred in particular areas in cases 2 and 3. This was because the given heat flux was applied across almost the entire free surface. Table 4 shows that the highest output temperature was reached at the outlet lip for case 3 at 2175 K, while similar temperatures were obtained for case 1 at 1966 K and case 2 at 1940 K. The temperature tended to be higher and there were larger temperature gradients on the free liquid surface when the scanning strategies were concentrated on certain locations.

The results in Figure 8b indicate that the liquid profile of the titanium alloy in the crucible mainly depended on the temperature field. In cases 1 and 4, the ratio of the liquid height to the hearth height remained at 14% across almost the entire hearth, while some of the alloy that was unaffected by the heat source solidified in cases 2 and 3.

### 4.2. The Effect of Scanning Strategy on the Al Evaporation Rate and Al Mass Fraction at the Outlet Lip

According to Langmuir’s equation, the evaporation rate of aluminum mainly depends on temperature and surface concentration. In order to predict the evaporation and Al mass fraction at the outlet, the mass transport through the top surface, the heat transfer, and the fluid flow were coupled together. The behavior of the Al mass fraction was examined from the inlet in order to keep track of it under the effect of diffusion in the molten flow. When there is no evaporation, the aluminum remains until it flows into the outlet. Under high temperatures, continuously irritated by the electron beam, the Al evaporates due to its high vapor pressure. As EBCHM operates under a vacuum environment, the activity coefficient, which can be determined through experimentation, plays a vital role in identifying the evaporation rate.

In the EBCHM process, the Al mass fraction can be investigated by controlling the scanning strategies. Figure 9 reveals that the scanning strategies in cases 1 and 4 caused similar trends in the Al distribution on the free surface. However, some areas in case 3 remained at their initial values due to the solidification problem. Regarding the effects of the scanning strategies, Figure 10 shows the temperature and Al mass fraction information extracted at the top centerline. It is apparent that both the temperature and Al distribution were similar: the temperature was fairly uniformly distributed at 2300 K across almost the entire top area. The elliptical scanning pattern had little effect on the temperature field (case 4 in Figure 8a) because the heat flux was mostly concentrated in the main scanning profile. Therefore, it had a weak effect on the Al mass fraction field. Furthermore, the proportion of Al gradually decreased from the initial value of 7.8% to 5.95% at the inlet and 5.86% at the outlet (Table 4). In case 2, the highest temperature (~2600 K) was obtained near the inlet and caused the largest evaporation at this area. Therefore, the Al mass fraction significantly decreased, by 0.4 m in terms of the hearth length, whereas a part of the surface temperature near the inlet in case 3 was low (~1800 K) due to the fact that no heat flux was applied in this area. This resulted in the solidification problem; the Al mass fraction could not evaporate with the constant value shown in Figure 10b. This solidification can cause severe problems, such as turbulent flow, the Marangoni effect, and unsteady temperatures and Al mass fractions at the outlet. Therefore, the scanning strategies with a uniform heat flux distribution across the full hearth could enhance the quality of the liquid alloy before it began to solidify.

## 5. Conclusions

In this study, four different electron beam scanning strategies were applied to investigate their effects on the Al mass fraction field related to evaporation under low pressure in a water-cooled hearth. A coupled thermal-flow model was developed based on the physical phenomena that occur in the EBCHM process. A 2D numerical model based on the electron beam button melting (EBBM) problem was validated with the experimental data from Zhongkui Zhang’s work [14], and it was confirmed that the model was acceptable for the prediction of the effects of scanning strategies on the Al evaporation of a titanium alloy in the EBCHM process. The conclusions are as follows:The heat flux of each of the scanning strategies could be calculated using the time-averaged heat-flux approach;The temperature distribution was successfully predicted with the different scanning strategies. At an inlet flow rate of 400 kg/h, the full-hearth heat flux (case 1) and the elliptical-pattern heat flux (case 4) provided the most uniform temperatures on the free surface and resulted in no significant changes in the Al mass fraction. Therefore, the temperature and the proportion of the Al remained nearly steady;The melting pool profile was determined using the temperature distribution. The scanning strategies that were only applied to particular areas (cases 2 and 3) caused large deviations in the temperature distribution on the free surface, as well as a solidification problem due to the lack of heat flux to maintain the liquid phase;During the EBCHM process, the refining stage caused significant losses of Al mass through its evaporation. The numerical simulation showed potential for the prediction of Al loss in practice.

Overall, the present model successfully illustrated the effects of scanning strategies on Al evaporation in the EBHCM process. The obtained results demonstrated that the significant losses of Al in the refining stage can be controlled with appropriate scanning strategies. As recommendations for future work, the melting stage where there is a significant loss of Al in the initial phase change should also be considered. Secondly, an experimental investigation of several scanning strategies with different process parameters needs to be performed in order to clearly validate the results predicted in this study.

## Figures and Tables

**Figure 1 materials-15-00820-f001:**
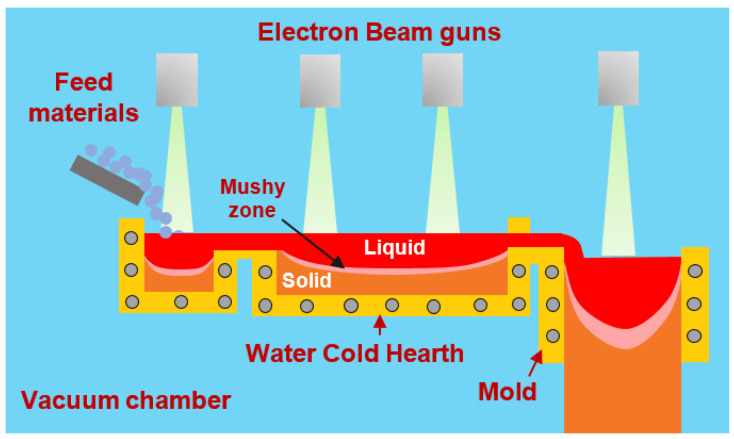
Schematic of the EBCHM process.

**Figure 2 materials-15-00820-f002:**
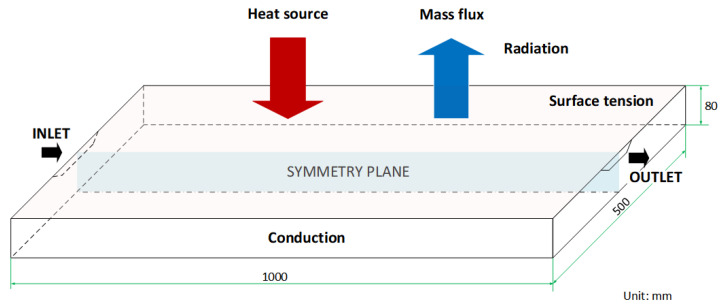
The geometry and boundary conditions of the numerical domain.

**Figure 3 materials-15-00820-f003:**
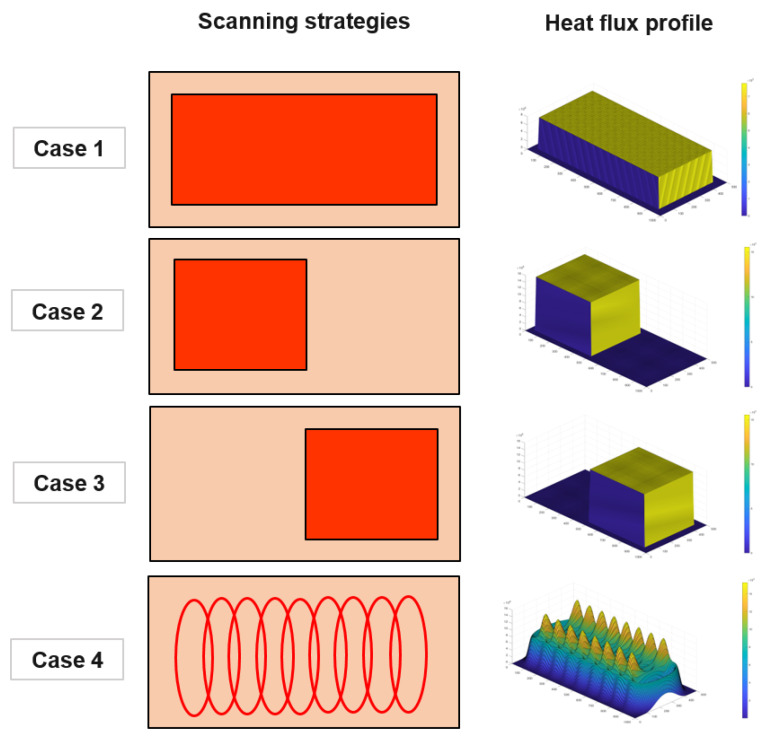
The four scanning strategies.

**Figure 4 materials-15-00820-f004:**
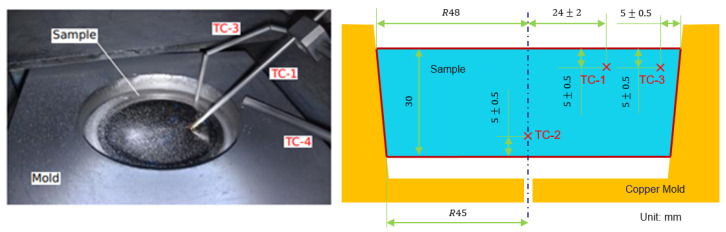
Experimental setup and schematic of the locations of the thermocouples in the EBBM model [14].

**Figure 5 materials-15-00820-f005:**
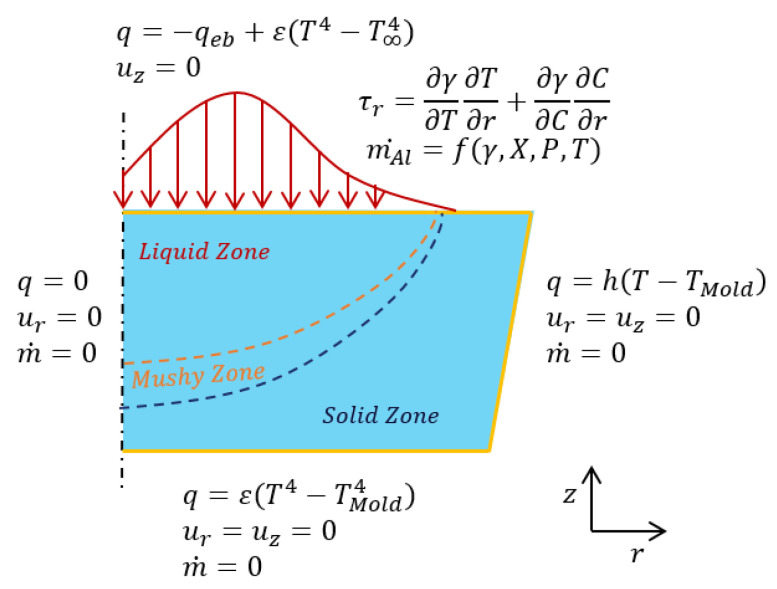
Summary of validation model boundary conditions.

**Figure 6 materials-15-00820-f006:**
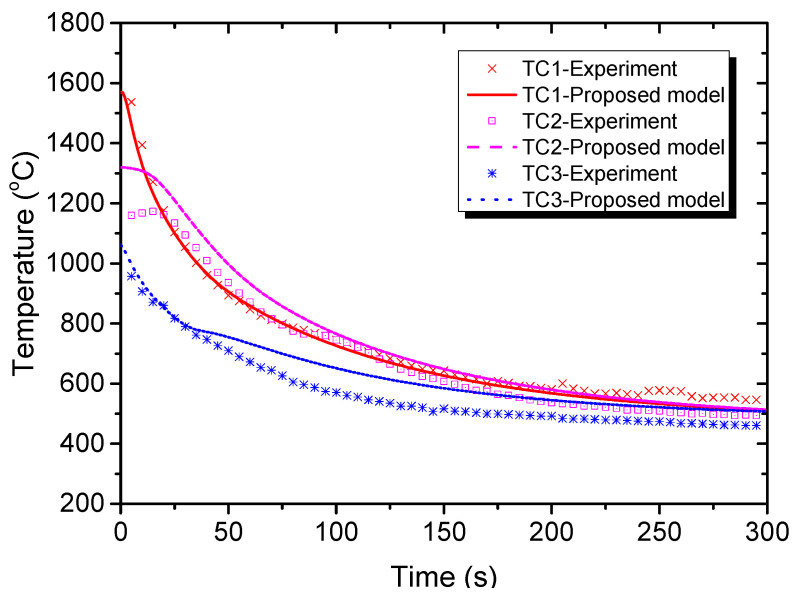
Temperature variation at three locations in the EBBM sample. TC = thermocouple.

**Figure 7 materials-15-00820-f007:**
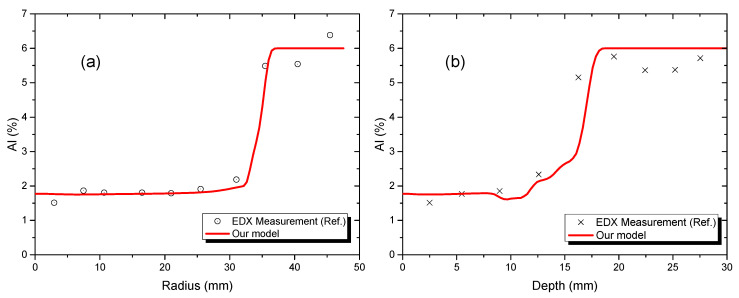
Comparison of predicted and measured Al concentration profiles in the EBBM sample. (**a**) Radius direction, (**b**) depth direction.

**Figure 8 materials-15-00820-f008:**
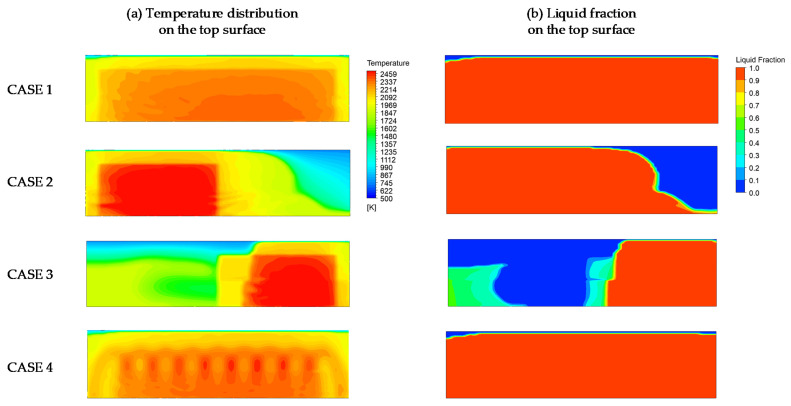
(**a**) Temperature fields and (**b**) liquid fractions under the four scanning strategies in the water-cooled hearth.

**Figure 9 materials-15-00820-f009:**
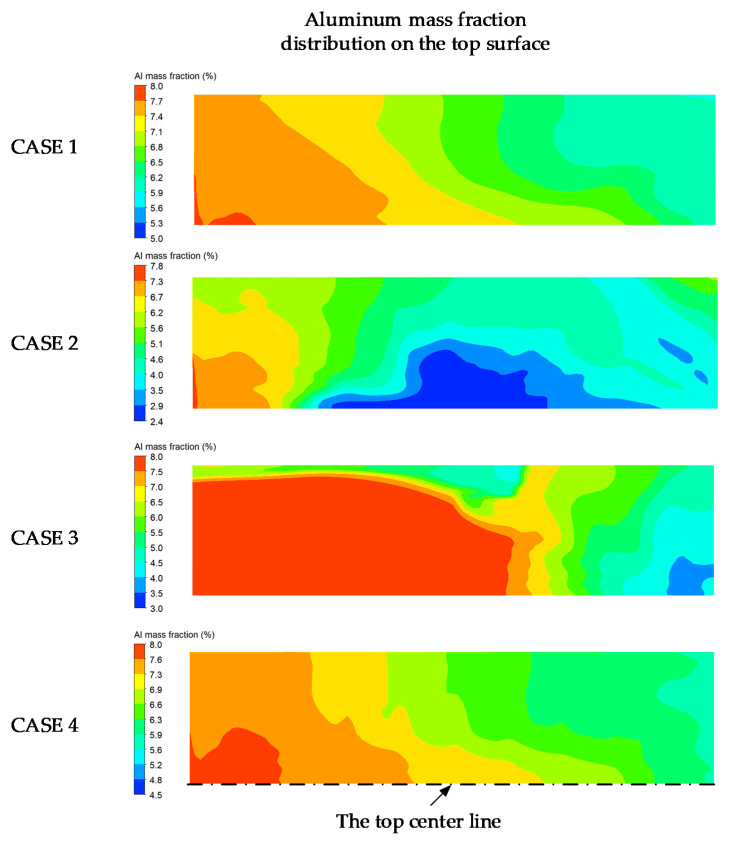
Al mass fraction fields with respect to the four scanning strategies after 550 s.

**Figure 10 materials-15-00820-f010:**
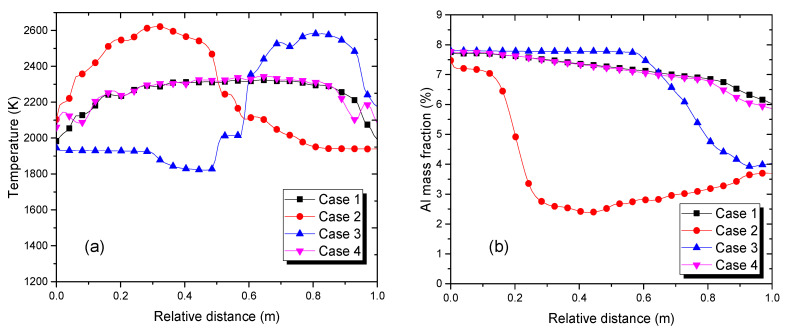
(**a**) The temperatures and (**b**) Al mass fractions for the top centerline.

**Table 1 materials-15-00820-t001:** Ti-6Al-4V properties.

Parameter	Unit	Value
Density	$\mathrm{kg}/m^{3}$	4110
Thermal conductivity	W/mK	37
Specific heat	J/(kgK)	794.2
Liquidus temperature	K	1946.5
Solidus temperature	K	1926.5
Emissivity	-	0.23
Viscosity	kg/ms	0.00052

**Table 2 materials-15-00820-t002:** Summary of electron beam scanning parameters.

Parameter	Unit	Value
Power, $P_{EB}$	kW	800
Absorption coefficient, $\eta_{EB}$	-	0.7
Electron beam radius, σ	mm	20
Ellipse profile, a, b		
Width, a	mm	7.5
Height, b	mm	17.5
Number of points in each ellipse profile, N	-	100

**Table 3 materials-15-00820-t003:** Summary of main parameters in EEBM experiments [14].

Parameter	Unit	Value
Electron beam power	kW	13.4
Temperature of the water-cooled furnace chamber, $T_{\infty }$	°C	15
Temperature of the water-cooled mold at the bottom surface	°C	50
Temperature of the mold wall	°C	200
Surface tension coefficient of temperature ∂γ/∂T	-	−4.5
Surface tension coefficient of concentration ∂γ/∂C	-	−0.16

**Table 4 materials-15-00820-t004:** Temperatures and Al mass fractions at the outlet lip after 550 s.

Case	1	2	3	4
Outlet temperature (K)	1966.94	1940.85	2175.33	2081.31
Outlet Al mass fraction (%)	5.95	3.69	4.03	5.86

## Data Availability

All data are available within the manuscript.

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
