# Peer review of "Numerical Simulation of the Effects of Scanning Strategies on the Aluminum Evaporation of Titanium Alloy in the Electron Beam Cold Hearth Melting Process"

_materials, 2022, doi:10.3390/ma15030820_

Round 1

Reviewer 1 Report

Find enclosed my report.

Reviewer 2 Report

The work is very interesting. However, some details are required before the acceptance of the current manuscript. Also, some information is misleading. Thus, a major revision is required.

  1. More details for the model setting should be provided, including the UDF attached, the equations that describes the moving of the heat source, the settings of the model including the discretization models and other settings for the model.
  2. The solidification equation, namely Eq.(6), is not correct if the species transport model is activated.
  3. The grid-independent converged solutions should be provided.
  4. The figure qualities should be improved.
